# Minimum 2-Year Radiographic and Clinical Outcomes of Unrestricted Kinematic Alignment Total Knee Arthroplasty in Patients with Excessive Varus of the Tibia Component

**DOI:** 10.3390/jpm12081206

**Published:** 2022-07-25

**Authors:** Yaron Bar Ziv, Ahmad Essa, Konstantin Lamykin, Najib Chacar, Gilad Livshits, Salah Khatib, Yoav Comaya, Noam Shohat

**Affiliations:** Assaf Harofeh Medical Center, Sackler Medical School, Tel Aviv University, Ramat Aviv 69978, Israel; tawohemiz@gmail.com (A.E.); klamykin@gmail.com (K.L.); najib.chacar@gmail.com (N.C.); greliv13@gmail.com (G.L.); salah.khateeb@gmail.com (S.K.); yoavcomay6@gmail.com (Y.C.); noam.stam2@gmail.com (N.S.)

**Keywords:** kinematic alignment, varus, tibia angle, reported outcomes, arthroplasty

## Abstract

Kinematic alignment (KA) total knee arthroplasty (TKA) has gained much attention in recent years. However, debate remains on whether restrictions should be made on the tibia cut. The purpose of this study was to assess the safety and functional outcomes of excessive varus cuts. A single-center, retrospective analysis of consecutive patients undergoing TKA between 2018 and 2020 who had a minimum 2-year follow-up was conducted. EOS™ imaging conducted before and after surgery was analyzed for overall alignment, as well as for tibia and femur component positioning on the coronal planes. Patients were interviewed and asked to fill several questionnaires, including the visual analog score, Oxford knee score, and knee injury and osteoarthritis outcome score. Overall, 243 patients (71.9%) had a coronal tibial plate angle under 5° (moderate) and 95 patients (28.1%) had an angle above 5° (excessive). There were no significant differences between the moderate and excessive groups in patient-reported outcomes, nor were there differences in the number of patients achieving the minimal clinical difference. There were no cases of catastrophic failure or loosening. Unrestricted KA and excessive varus of the tibial component appears to be safe and efficient in relieving pain and restoring function for a minimum of 2 years following surgery.

## 1. Introduction

Symptomatic knee osteoarthritis (OA) affects 6% of the adult population and occurs in 10% of adults over age 60 years [1]. Several mechanisms have been proposed to play a role in its development [2]. The condition is progressive and results in the loss of the articular cartilage, and often leads to end-stage arthritis of the knee. Treatment includes nonoperative measures in the early stage [3,4]. However, when conservative treatment fails, total knee arthroplasty (TKA) is often required to alleviate pain and restore functionality [5]. The number of TKAs performed annually continues to rise [6]. While these surgeries are considered to be very successful overall in relieving pain and improving function, a large percentage of patients are unhappy with their artificial knee [7,8,9].

In an effort to increase satisfaction from TKA, the last decade has brought about advancements in the technique and technology [10]. There has been an increased interest in alternative alignment techniques to the conventional mechanical axis (MA), with kinematic alignment (KA) showing promising results in terms of satisfaction, functionality and safety [11,12,13,14]. While there is accumulating literature to support the superiority of calipered KA over conventional MA, there are still disagreements about whether the tibia cut includes the proper technique and if there is a need to set boundaries to avoid excessive varus (i.e., restricted KA) and early complications, such as the loosening of the tibia tray [15,16,17].

Another major change in TKAs in the last decade was the incorporation of new technologies, mainly robotic-arm-assisted surgeries [18]. The KA literature and other publications show that it is safe for the tibia tray to slightly deviate from being perpendicular to the tibia mechanical axis, which permits the standard robotic tibia cut to allow for a 3-degree play [19,20,21]. However, as with restricted KA, robotic arm manufacturers and surgeons fear tibia angles that rise above those boundaries.

During the past 5 years, our institution transitioned to caliper-based unrestricted KA using the linked technique in which the femur and soft tissue guide the tibia cut [22]. Consequently, our tibia cut is performed without any restraints. The aim of this study was to assess the medium-term safety and functionality of our technique and to directly compare the outcomes of patients with a tibia cut above and below what is considered to be excessive.

## 2. Materials and Methods

This was a retrospective, single-center study preformed between January 2018 and March 2020 to allow for a minimum 2-year follow-up. Following IRB approval, all primary total knee arthroplasty (TKA) cases performed by 3 fellowship-trained surgeons were queried from the hospital electronic records. Revision cases and valgus knee arthroplasty cases (*n* = 51) were excluded from this study. Electronic medical records were queried for the patient age, body mass index (BMI), comorbidities (using the Charlson comorbidity index), type of anesthesia (spinal versus general), operative time, and length of stay (LOS).

### 2.1. Technique

Starting in January 2018, our institution transitioned from mechanical axis (MA)-based TKA to calipered kinematic alignment (KA) using the linked technique, which was previously described in detail [22]. In short, the technique involves resurfacing the femur using the conventional calipered technique, which thereafter serves as a guide to cutting the tibia. Shims are used to distract the tibia and achieve soft tissue balance, thus sparing the need for medial release and avoiding cutting into soft bone. All surgeries were performed with a medial pivot knee design of the same manufacturer. No stems or constrained implants were used.

### 2.2. Radiographic Analysis

Standard protocol at our institution includes EOS imaging at preadmission testing (2–3 weeks before surgery) as well as at the first follow-up visit two weeks after discharge (Figure 1). Various measurements were taken, including the medial proximal tibia angle (MPTA), lateral distal femoral angle (LDFA), hip knee angle (HKA), and tibial slope. Tibia bone resorption (TBR) was also measured when plain X-rays were available [23,24]. Radiographic analysis was performed by 3 orthopedic residents (AE, SH, and GL), who were blinded to the clinical outcome assessment. To confirm interobserver reliability, 20 overlapping cases were examined showing a correlation (kappa) of 0.88 (95% confidence interval, 0.79 to 0.96).

### 2.3. Follow-Up Examination

All patients operated on during the above-mentioned period were contacted and invited to visit the clinic. Those who were not able to attend were interviewed over the phone by 3 medical students. Patients were asked to complete a visual analog score (VAS), Oxford knee score (OKS), and the knee injury and osteoarthritis outcome score (KOOS). Minimal clinical differences for OKS and KOOS were based on the prior literature [25,26]. Patients were also asked about readmissions and reoperations associated with the operated joint. Range of motion was documented at the most recent clinical visit.

### 2.4. Statistical Analysis

Descriptive statistics were calculated for all background characteristics and univariable analysis was conducted using a chi-square test for nominal data. Interval data were analysed by a t-test for normally distributed data (determined by the Kolmogorov–Smirnoff test) or Mann–Whitney U test (if not normally distributed). The interclass coefficients (kappa) were calculated to evaluate the reliability and reproducibility between and within readers. All analysis was performed using the SPSS packages (version 28.0.1,Armonk, NY: IBM Corp). Tibia tray angles on the coronal plane were grouped into two categories: moderate (MPTA between 85 and 90°); and excessive (MPTA below 85°). Comparisons were made between the two groups. A t-test was used to compare continuous variables, and a chi-square test was used for comparing categorical variables.

## 3. Results

Of the 385 patients who underwent TKA during the study period and met inclusion criteria, 8 patients died and 39 patients refused to participate, resulting in a total of 338 patients who were included in the study. Of these 338 patients, 243 had an MPTA between 85° and 90° (moderate), and 95 patients had an MPTA below 85° (excessive). Time to follow-up was 3.43 years (SD 0.79) in the moderate group, compared with 3.05 years (SD 0.73) in the excessive group (*p* < 0.001). Other than that, there were no differences between the two groups in terms of their baseline demographics, comorbidities, or range of motion. The two groups also shared similarities in pain and functionality prior to the surgery (Table 1).

Patients in the moderate group had a smaller mean MPTA (85.76°, SD 3.38°) prior to surgery compared with patients in the excessive group (83.67°, SD 3.35°) (*p* < 0.001). The average HKA was also smaller in the moderate group (9.47°, SD 4.59°) than in the excessive group (11.31°, SD 5.95°) (*p* < 0.002). These differences were even more pronounced after surgery. The mean MPTA and HKA changed to 87.96° (SD 2.07°) and 1.99° (SD 3.45°), respectively, in the moderate group. The mean MPTA and HKA changed to 82.63° (SD 1.83°) and 3.13° (SD 2.57°), respectively, in the excessive group (Table 2). Postoperatively, there were also differences in the LDFA between the two groups; the mean angle was 85.22° (SD 4.16°) in the moderate group compared with 83.34° (SD 3.79°) in the excessive group (*p* < 0.001).

In both groups, a significant improvement in pain and function was seen following surgery (*p* < 0.001). There were no significant differences between the moderate and excessive groups in average VAS scores, OKS, or KOOS (Figure 2), nor were there differences in the number of patients achieving MCID (Table 3). There were also no significant differences in the range of motion between the two groups; the mean extension and flexion ranged between 2.34° (SD 4.1°7) and 114.86° (SD 14.32°) in the moderate group, compared with between 1.40° (SD 3.06°) and 113.20° (SD 15.73°) in the excessive group (*p* = 0.291 and 0.608, respectively).

During the study period, two patients required a reoperation: one from the moderate group and one from the excessive group (0.4% vs. 1.1%, *p* = 0.484). One revision was for a periprosthetic joint infection and the other revision was for a patellar dislocation. There were no cases of aseptic loosening, and the TBR was 3.26 mm (3.25) in the moderate group compared with 1.6 mm (2.20) in the excessive group (*p* = 0.089).

## 4. Discussion

The aim of this study was to assess the safety of extreme varus positioning of the tibial component in TKAs. To the best our knowledge, this is the largest study to date that included patients with an MPTA smaller than 5° (excessive varus). The main findings were that excessive varus did not lead to increased failure at a mean 3.05 years from surgery, nor did it lead to increased pain or impaired function as reflected by patient-reported outcome scores.

The debate on whether mechanical and neutral tibial alignment is associated with superior durability and survivorship following TKA continues to garner interest as alternative alignment techniques show functional advantages [21,27,28,29,30,31,32,33]. One recent long-term follow-up study (>10 years) supported the safety of alignment outside of the mechanical range (0 ± 3°); however, the tibia component varus was not specifically assessed [21]. Howell et al. compared patients with a neutral (≤0°) and varus (>0°) tibial component alignment and found similar survivorship and patient-reported outcomes [29,34]. Another recent matched control study (*n* = 66) compared long-term outcomes of neutral alignment (90 ± 3°) and varus alignment (<87°) based on tibial coronal alignment (PTA) and found similar results [30]. However, none of the aforementioned studies specifically examined coronal tibial alignment above 5°, and the total number of patients outside of the 5° range within previous cohorts is extremely limited.

Our calipered, KA-based surgical technique relies on linking the femur to the tibia to perform the tibial cut, thus avoiding recutting the tibia, but on occasion resulting in extreme varus positioning of the tibia. This resulted in the largest cohort to date with so-called “extreme outliers”, as 95/338 (28.1%) of our cohort had a tibial varus alignment above 5°. The fact that none of these patients had a catastrophic failure or early loosening reinforces the results presented in the aforementioned studies in this relatively extreme patient population. While preliminary, our results support not only the safety of KA, but also specifically the safety of unrestricted KA. Restricted KA is performed so that the tibial and femoral cuts are always kept within 5° of the mechanical axis, and the HKA must always fall within 3° of neutral. The technique was developed to avoid the restoration of extreme anatomies, which may result in diminished survivorship due to inappropriate implant designs or fixation methods. Supporters of restricted KA may also fear that some knee anatomies may be biomechanically inferior. The results, while limited in the length of follow-up time, support the medium-term safety and adequate functionality of unrestricted KA.

This study is not without limitations. While it was limited to patients at least 2 years from surgery, it was still a relatively short time span for follow-up and longer follow-up time spans are needed. However, the fact that we did not witness any catastrophic failure or early loosening signs on X-rays is reassuring. Furthermore, all surgeons were fellowship-trained and experienced in the caliper-based surgical technique, which aims to restore joint line obliquity and the prearthritic state. We cannot support the safety of an extreme tibial component varus when using other surgical techniques. In addition, all surgeries were performed using a medial pivot design of a single manufacturer, and while it reasonable that prosthesis of other manufacturers will be just as safe, this must be examined in the future. Finally, all patients within the cohort received unrestricted KA surgery, so we could not directly compare the functional results with those of other alignment techniques.

In conclusion, unrestricted KA and excessive varus of the tibial component appears to be safe and efficient in relieving pain and restoring function at a minimum of 2 years following surgery. Further long-term follow-up is in place.

## Figures and Tables

**Figure 1 jpm-12-01206-f001:**
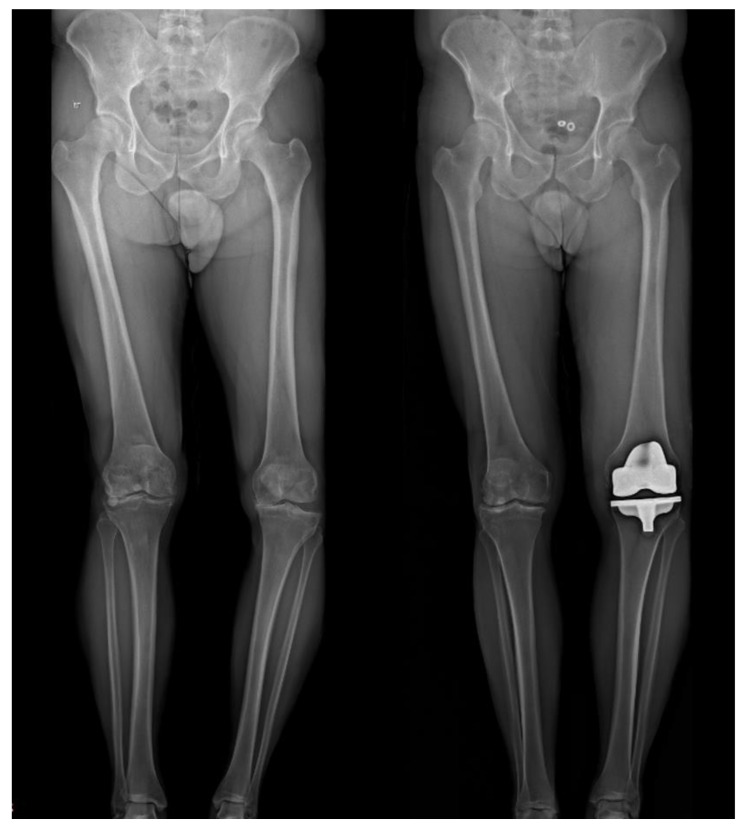
A 46-year-old man with a postoperative MPTA of 78°, LDFA of 86°, and HKA of 8.6°. Preoperative VAS was 6, OKS was 11, and overall KOOS was 43 (symptoms 53, pain 63, function 26, and QOL 27). Postoperative scores improved to a VAS of 0, OKS of 45, and overall KOOS of 93.2 (symptoms 92.86, pain 100, function 92.65, and QOL 81.25) at 4.3 years following surgery.

**Figure 2 jpm-12-01206-f002:**
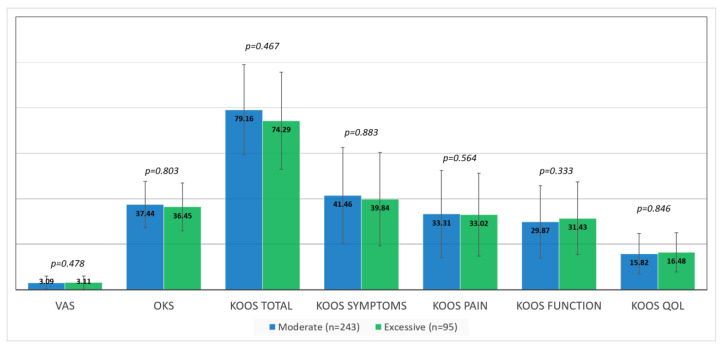
Average patient-reported outcome scores in the moderate versus excessive groups. (KOOS) knee injury and osteoarthritis outcome *Score*; (OKS) Oxford knee score; (VAS) visual analog scale; and (QOL) quality of life.

**Table 1 jpm-12-01206-t001:** Baseline characteristics, operative factors, and patient-reported outcomes in the moderate versus excessive groups.

Variable	Moderate (*n* = 243)	Excessive (*n* = 95)	*p*-Value
Age	70.16 (8.43)	70.83 (8.12)	0.221
Sex (female)	164 (67.5%)	57 (60.0%)	0.205
BMI (kg/m^2^)	31.65 (5.03)	31.27 (5.48)	0.699
CCI	0.85 (1.10)	0.712 (1.06)	0.674
Anesthesia (spinal)	180 (74.1%)	65 (68.4%)	0.343
Operative duration	82.39 (20.35)	84.77 (21.76)	0.238
LOS	4.32 (2.69)	4.49 (10.21)	0.105
Extension	4.11 (5.21)	4.42 (5.71)	0.785
Flexion	109.08 (15.94)	111.54 (16.23)	0.483
VAS	8.07 (1.43)	8.05 (1.47)	0.969
OKS	13.78 (7.75)	13.45 (7.64)	0.283
KOOS TOTAL	28.22 (15.22)	30.52 (14.31)	0.606
Time to Follow Up (m)	41.25 (9.52)	36.65 (8.77)	<0.001

(BMI) bone mass index; (CCI) Charlson comorbidity index; (LOS) length of stay; (KOOS) knee injury and osteoarthritis outcome score; (OKS) Oxford knee score; (VAS) visual analog scale; (m) months.

**Table 2 jpm-12-01206-t002:** Preoperative and postoperative alignment in the moderate and excessive groups.

	Preoperative			Postoperative		
	Moderate (*n* = 243)	Excessive (*n* = 95)	*p*-Value	Moderate (*n* = 243)	Excessive (*n* = 95)	*p*-Value
MPTA	85.76° (3.38°)	83.67° (3.35°)	<0.001	87.96° (2.07°)	82.63° (1.83°)	<0.001
LDFA	89.36° (3.87°)	89.69° (4.12°)	0.49	85.22° (4.16°)	83.34° (3.79°)	<0.001
HKA *	−9.47° (4.59°)	−11.31° (5.95°)	0.002	−1.99° (3.4°)	−3.13° (2.57°)	0.010
Slope	10.68° (5.34°)	11.41° (6.31°)	0.058	7.17° (4.08°)	6.89° (3.58°)	0.553

MPTA (medial proximal tibial angle); LDFA (lateral distal femoral angle); HKA (hip knee angle); * negative numbers represent varus.

**Table 3 jpm-12-01206-t003:** Number and percentage of patients achieving minimal clinical differences (MCID) of the Oxford knee score (OKS) and the knee injury and osteoarthritis outcome Score (KOOS) subcategories in the moderate versus excessive groups.

MCID	Moderate (*n* = 243)	Excessive (*n* = 95)	*p*-Value
OKS	228 (93.8%)	93 (98.4%)	0.287
KOOS Symptoms	204 (84.0%)	78 (82.0%)	0.691
KOOS Pain	178 (73.1%)	79 (83.6%)	0.114
KOOS Function	207 (85.3%)	79 (83.6%)	0.834
KOOS QOL	215 (88.5%)	86 (90.2%)	0.814

(KOOS) knee injury and osteoarthritis outcome score; (OKS) Oxford knee score; (QOL) quality of life.

## Data Availability

The data presented in this study are available on request from the corresponding author. The data are not publicly available due to patient privacy reasons.

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
