# Peer review of "Minimum 2-Year Radiographic and Clinical Outcomes of Unrestricted Kinematic Alignment Total Knee Arthroplasty in Patients with Excessive Varus of the Tibia Component"

_jpm, 2022, doi:10.3390/jpm12081206_

Round 1

Reviewer 1 Report

 The authors have developed an interesting and useful retrospective study of consecutive patients undergoing TKA  to assess the safety and functional outcomes of excessive varus cuts.

However, I would like to make some observations before recommending your work for publication.

  1. Please write the title correctly and with PICO in mind.
  2. EOS acronyms should be defined in the Abstract.
  3. The authors should further develop the Abstract, mentioning the causes of osteoarthritis, which is the main cause leading to TKR. I recommend them to comment on the following work, as a multisystemic origin of osteoarthritis: doi:10.3390/nu13030716.
  4. I recommend the authors to comment briefly in the Introduction on the conservative treatment of osteoarthritis of the knee, such as rehabilitation. I recommend the following references of studies investigating the use of dry needling and a 3-month program of therapeutic exercise, and manual therapy: doi:10.1093/pm/pnz036 and doi:10.3390/app11041895.
  5. The Discussion part should be further developed. Since claims are made in your study about safety, I recommend the authors comment the following paper on the impact of knee TKR from a systemic point of view: DOI: 10.1097/TGR.0000000000000337

Reviewer 2 Report

Dear authors. Thanks for considering me for JPM review.

The topic is highly debated in these last years and you have focused your efforts on one of the major subjects of detrimental comments about unrestricted KA: the varus of the tibia beyond the safe zone. 

I really appreciate your efforts. the paper is well written and easy to understand, the limitations are clear for readers. Although the follow up is midterm the absence of early or catastrophic failures draws a picture of general safety for the patients. 

Author Response

Thank you for reviewing our paper. 

Round 2

Reviewer 1 Report

The authors have substantially improved their work. I recommend their work for publication.

Congrats